# Functional *MICA* Variants Are Differentially Associated with Immune-Mediated Inflammatory Diseases

**DOI:** 10.3390/ijms25053036

**Published:** 2024-03-06

**Authors:** Chin-Man Wang, Keng-Poo Tan, Yeong-Jian Jan Wu, Jian-Wen Zheng, Jianming Wu, Ji-Yih Chen

**Affiliations:** 1Department of Rehabilitation, Chang Gung Memorial Hospital, Chang Gung University College of Medicine, Taoyuan 33302, Taiwan; cmw1314@cgmh.org.tw; 2Department of Medicine, Division of Allergy, Immunology and Rheumatology, Chang Gung Memorial Hospital, Chang Gung University College of Medicine, Taoyuan 33305, Taiwan; helentan.tw@yahoo.com.tw (K.-P.T.); yjwu1962@gmail.com (Y.-J.J.W.); pqr780925@cgmh.org.tw (J.-W.Z.); 3Department of Veterinary and Biomedical Sciences, Department of Medicine, University of Minnesota, St. Paul, MN 55108, USA

**Keywords:** MICA alleles, NKG2D, psoriasis, rheumatoid arthritis, systemic lupus erythematosus

## Abstract

As the principal ligand for NKG2D, MICA elicits the recruitment of subsets of T cells and NK cells in innate immunity. MICA gene variants greatly impact the functionality and expression of MICA in humans. The current study evaluated whether *MICA* polymorphisms distinctively influence the pathogenesis of psoriasis (PSO), rheumatoid arthritis (RA), and systemic lupus erythematosus (SLE) in Taiwanese subjects. The distributions of *MICA* alleles and levels of serum soluble NKG2D were compared between healthy controls and patients with PSO, RA, and SLE, respectively. The binding capacities and cell surface densities of MICA alleles were assessed by utilizing stable cell lines expressing four prominent Taiwanese *MICA* alleles. Our data revealed that *MICA**010 was significantly associated with risks for PSO and RA (P_FDR_ = 1.93 × 10^−15^ and 0.00112, respectively), while *MICA**045 was significantly associated with predisposition to SLE (P_FDR_ = 0.0002). On the other hand, *MICA**002 was associated with protection against RA development (P_FDR_ = 4.16 × 10^−6^), while *MICA**009 was associated with a low risk for PSO (P_FDR_ = 0.0058). *MICA**002 exhibited the highest binding affinity for NKG2D compared to the other *MICA* alleles. Serum concentrations of soluble MICA were significantly elevated in SLE patients compared to healthy controls (*p* = 0.01). The lack of cell surface expression of the *MICA**010 was caused by its entrapment in the endoplasmic reticulum. As a prevalent risk factor for PSO and RA, *MICA**010 is deficient in cell surface expression and is unable to interact with NKG2D. Our study suggests that *MICA* alleles distinctively contribute to the pathogenesis of PSO, RA, and SLE in Taiwanese people.

## 1. Introduction

Natural killer (NK) cells possess an extensive repertoire of activating and inhibiting receptors, comprising natural killer group 2 member D (NKG2D), NKp30, NKp44, NKp46, killer-cell immunoglobulin-like receptors (KIR), and CD94/NKG2A receptors. NK cells employ inhibitory and activating receptors to monitor alterations in the expression of human leukocyte antigen (HLA) class I molecules and stress-induced ligands. This surveillance enables NK cells to achieve functional proficiency while gaining authorization for the elimination of target cells [1]. NK cells exert their impact across the innate and adaptive immune systems via their cytotoxicity and the secretion of cytokines. Throughout the process of immune surveillance, cytotoxic NK cells are tasked with promptly eradicating virus-infected or altered tumor cells. NK cells possess the ability to regulate immune responses through interaction with several cell types, including dendritic cells, macrophages, T cells, and endothelial cells. These interactions can either amplify or suppress immune responses [2]. There is increasing evidence indicating that NK cells, when acting as effector cells, can either promote the development of inflammation and autoimmunity or serve as a protective mechanism against their own detrimental effects [3,4]. The dysregulation of NK cells has the potential to result in an overactive inflammatory response and the emergence of autoimmune diseases [4,5]. The major histocompatibility complex class I chain-related gene A (MICA) exerts a pivotal role in orchestrating the immune response [6]. MICA serves as a significant ligand for NKG2D, facilitating the activation of NK cells and promoting reciprocal stimulation with specific subsets of T cells [7]. The production of the MICA protein is induced by DNA damage and physicochemical stressors, leading to the activation and surveillance of lymphocytes with specialized immune-related tasks. MICA enhances the capacity of the immune system to recognize and eliminate infections through the activation of NK and CD8+ T cells [6,8,9]. The emergence of graft-versus-host disease and systemic lupus erythematosus (SLE) is attributed to the processes of NK cell-mediated cytotoxicity and the production of immunosuppressive soluble MICA (sMICA) particles [10,11,12]. Autoimmune inflammatory illnesses emerge as a result of an imbalance in the regulation of immunological activation and tolerance. A recent study found that specific variants of MICA SNP haplotypes were strongly linked to a higher vulnerability to ankylosing spondylitis [13]. This finding implies MICA plays an essential part in the onset of autoimmune inflammatory illnesses. Additionally, a growing list of evidence indicates that NKG2D-positive CD4+ T cells and NK cells contribute to the pathogenesis of SLE and RA through interactions with MICA and sMICA/B [14]. Accordingly, NKG2D and NKG2D-L are important therapeutic targets in the treatment of autoimmune disorders by blocking the interaction between NKG2D of immune cells and MICA on non-immune cells. The current study sought to further evaluate the impact of *MICA* variants on the development of psoriasis (PSO), RA, and SLE. The data gathered in our study show that *MICA**010 is a prevalent risk factor for both PSO and RA, while *MICA**045 is a significant risk factor for SLE specifically in the Taiwanese population. The implications of our findings suggest that distinct *MICA* variants play differential roles in the development of immune-mediated inflammatory disorders (IMIDs).

## 2. Results

### 2.1. MICA*010 Is a Common Risk Factor for PSO and RA, Whereas MICA*045 Is Associated with SLE Susceptibility

In healthy Taiwanese controls, the allele frequencies of seven *MICA* alleles (*MICA**010.01, *008.01, *002.01, *019.01, *012.01, *045, and *009.01) were greater than 0.05 (or 5%) and could be regarded as population-specific common alleles (Table 1). In the present genetic association investigation, we focused on those 7 common *MICA* alleles formed by 17 *MICA* non-synonymous coding SNPs (Appendix A). We found that *MICA**010.01 was significantly associated with the risk for PSO (P_FDR_ = 1.93 × 10^−15^; OR = 2.18; 95% CI = 1.82–2.62) (Table 1) and RA (P_FDR_ = 0.0011; OR = 1.51; 95% CI = 1.18–1.93) (Table 2), suggesting that *MICA**010.01 is the common risk factor for PSO and RA. In contrast, *MICA**009.01 was significantly associated with protection against the development of PSO (P_FDR_ = 0.0058; OR = 0.53; 95% CI = 0.38–0.75) (Table 1), and *MICA**002.01 was significantly associated with protection against RA development (P_FDR_ = 4.16 × 10^−6^; OR = 0.53; 95% CI = 0.41–0.70) (Table 2), respectively. In addition to *MICA**010.01, *MICA**012.01 appears to confer RA susceptibility (P_FDR_ = 0.0330; OR = 1.41; 95% CI = 1.03–1.93) (Table 2). Notably, we identified that *MICA**045 was the only allele substantially associated with SLE susceptibility (P_FDR_ = 0.0002; OR = 2.24; 95% CI = 1.47–3.43) (Table 3). Taken together, our data indicate that the various *MICA* alleles have distinct roles in the development of diverse autoimmune diseases in Taiwanese people.

### 2.2. MICA*010 Is Deficient in Cell Surface Expression

MICA is a type I cell surface membrane protein with three extracellular domains. Two monoclonal antibodies (mAb 159,227 and 6D4) react with cell surface MICA extracellular domains. mAb 159,227 recognizes an epitope specific for MICA, while mAb 6D4 binds to a common epitope presented on both MICA and MICB. As shown in Figure 1A, *MICA**008 and *MICA**019 can be detected by both mAbs. MICA*008 was expressed at the highest level, while the *MICA**019 expression level was intermediate among four alleles (*MICA**002, *008, *010, and *019). *MICA**002 was only detectable by mAb 6D4 but not by mAb 159227, suggesting that *MICA**002 may function similarly to MICB. Moreover, the expression level of *MICA**002 is lower than that of *MICA**008 and *MICA**019. Most importantly, both mAbs (159,227 and 6D4) were unable to detect *MICA**010 on the cell surface as fluorescent intensities of staining were equivalent to those of parental cells and vector control cells (Figure 1A).

Endo H removes high-mannose glycans from resident proteins in the endoplasmic reticulum (ER). It has been speculated that the absence of *MICA**010 on the cell surface may be caused by ER retention of peptides. To examine the trafficking of *MICA* variants, we performed cellular MICA digestion analyses with Endo H and PNGase F. As shown in Figure 1B, MICA*010 protein’s molecular weight changed from 60 kDa to 42 kDa after Endo H and PNGase F treatment, and the MICA*010 protein was sensitive to the treatment of both Endo H and PNGase F. The sensitivity of MICA*010 to Endo H indicates that the MICA*010 peptide is trapped in ER. On the other hand, Endo H treatment did not alter the electrophoretic mobility patterns of MICA*002, *019, and *008. As shown in Figure 1C, after PNGase F treatment, the molecule weights of MICA*008, MICA*002, and MICA*019 were reduced to 36, 42, and 42 kDa, respectively, indicating that MICA*008, MICA*002, and MICA*019 are effectively processed into highly glycosylated mature MICA for cell surface expression.

### 2.3. MICA Alleles Have Different Binding Capacity for NKG2D

MICA is a major NKG2D ligand. Interaction between MICA on target cell surface and NKG2D on NK cells leads to NK cell activation and subsequent cytotoxicity. We used a flow cytometry-based binding assay to determine the effects of *MICA* alleles on their ability to interact with NKG2D. Surface staining of MICA on the transfected LCL cells was also performed and used as a normalization control. As shown in Figure 2, the binding of soluble NKG2D-Fc to *MICA**002 was the highest as compared to all the other *MICA* alleles. The NKG2D binding capacity was similar between *MICA**008 and *MICA**019. *MICA**010 cells failed to bind NKG2D-Fc, which could be explained by the undetectable level of cell surface MICA*010. 

### 2.4. Soluble MICA (sMICA) Levels Were Significantly Increased in Serum Samples of SLE Patients

Next, we examined the levels of sMICA in patients with SLE, RA, and psoriasis, as well as healthy controls. As depicted in Figure 3, serum sMICA concentrations were significantly higher in SLE patients than in healthy controls (*p* = 0.01). On the other hand, serum sMICA levels were comparable in healthy controls and in RA and psoriasis patients. Of note, we were unable to detect sMICA in RA and psoriasis patients carrying the homozygous *MICA**010 allele, confirming that MICA*010 is a deficient allele for mature MICA expression. 

## 3. Discussion

MICA molecule engages C-type lectin activator receptor NKG2D on NK, γδ, and CD8^+^ αβ T cells to counteract the inhibiting response mediated by Killer Inhibitory Receptors (KIR) and/or CD94/NKG2A/B molecules [15]. The humoral response induced by MICA may contribute to graft rejection of solid organ transplantation [6,10,16,17,18]. MICA belongs to the non-classical class I family with highly polymorphic human stress antigens associated with tumor and inflammation surveillance [8]. A multitude of MICA functional variations have been discovered in global populations. The presence of transmembrane-encoded microsatellite triplet repeat polymorphism and coding sequence polymorphisms in MICA had a substantial impact on the expression of RNA and protein [8]. The *MICA*-A5.1 polymorphism results in the creation of a truncated protein that leads to greater amounts of sMICA in circulation. A substitution of valine with methionine at position 129 (*MICA*-129) of the MICA protein has been found to impact its capacity to bind to NKG2D, its cytotoxicity, interferon-γ release, and the density of its expression on the plasma membrane [8,19]. The *MICA* 129 methionine (Met) variation exhibited enhanced NKG2D signaling, eliciting greater NK-cell killing and interferon-γ release. Additionally, *MICA*-129Met induced rapid co-stimulation of CD8(+) T cells, resulting in a more speedy and pronounced reduction in NKG2D expression on both NK and CD8(+) T cells relative to the MICA-129 valine (Val) version [20]. 

Although over one hundred *MICA* alleles have been identified, their precise functional implications remain unknown. It is crucial to provide clarification concerning the unique distribution patterns of MICA alleles and the wide array of ethnic variations. We examined the impact of *MICA* alleles, which are formed by all coding SNPs, on the susceptibility to SLE, RA, and psoriasis in Taiwanese subjects. Specifically, *MICA**010 was identified as a common risk factor for psoriasis and RA in Taiwanese subjects. In contrast, *MICA**002, *MICA**009, and *MICA* *012 played a protective role against the development of psoriasis. We confirmed that *MICA**10 is incapable of producing a functional MICA in either soluble or membrane-bound form [13]. Furthermore, we demonstrate that the *MICA**10 peptide becomes entangled in the ER, preventing its expression on the cell surface and producing sMICA in the current study. We also found that cell surface *MICA**002 and *MICA**010 bound to NKG2D with significantly different degrees of avidness. Our data suggest that the deficiency of MICA, a critical immune-stimulatory ligand for NKG2D, is an underlying mechanism for the susceptibility to psoriasis and RA. Functional MICA alleles are distinctively associated with the pathogenesis of psoriasis and RA in Taiwanese subjects.

Psoriasis represents an intricate autoimmune disease wherein cellular and molecular inputs to an excess immune reaction are currently identified [21]. The major histocompatibility complex (MHC) encompasses the susceptible locus for psoriasis with the greatest impact magnitude, which is influenced to some extent by variations of the *HLA-Cw**0602 allele [22]. MICA is thought to be primarily stress-induced in psoriasis. Independent of *HLA-B* and *HLA-C*, the *MICA*-129Met allele, especially Met/Met homozygosity, had a strong correlation with the two types of cutaneous psoriasis (PsC) and psoriatic arthritis (PsA) [23]. The *MICA*-A9 variant, in accordance with the *MICA**002 allele, is a potential genetic marker for the occurrence of PsA, irrespective of the HLA-C association [24,25]. Psoriasis was associated with SNPs with potent immunological function, namely, rs2507971, rs9260313, rs66609536, and rs380924, located in the class I region of the MHC [22]. SNP rs13437088, situated at the 16 kb telomeric region of *MICA* and 30 kb centromeric region of *HLA-B,* exhibited a robust association with Han Chinese psoriasis after adjusting the effect of the imputed *HLA-Cw**0602 [26]. Furthermore, the prevalence of the MICA-A5.1 allele was substantially greater among Chinese subjects [27]. Risk alleles for *HLA-B* and *HLA-C* determine susceptibility to autoimmunity and inflammation. The *MICA**010-*HLA-B**4601-*Cw**01 haplotype displayed a greater incidence of Type I and II psoriasis. Significantly, *MICA**010 exhibited a robust correlation with *HLA-Cw**01 [28], which correspond to two prominent risk loci in the Asian population, including Taiwanese subjects [29,30]. 

RA was associated with *MICA*-129Val, and RA patients had elevated sMICA levels [31]. In addition, *MICA* polymorphisms are associated with *HLA-DRB1* shared epitope (SE) presence and autoantibody production in RA patients [32]. Support for the correlation of *MICA*-250 (rs1051794) with RA occurred independently of the recognized *HLA-DRB1* risk alleles, suggesting that *MICA* is a susceptible gene [33,34]. The *MICA* rs1051792 polymorphism modifies the therapeutic response of RA patients to TNF-blocking therapy [35]. In the Tunisian population, the *MICA*-A9 allele was substantially associated with RF-negative RA patients, while the severity of RA may be influenced by the *MICA*-TM and *MICA* 129met/val genotypes [33]. The correlation observed in Koreans between the *MICA*-A4 allele and RA was found to be associated with *HLA-DRB1**0405, whereas the *MICA*-A9 allele could possess a mild protective role toward RA susceptibility [36]. The *MICA*-A6 allele was an independent predictor of RA protection [37] in the SE+ subset of RA patients [38]. The functional confirmation of *MICA**010 as a risk factor was achieved through deficient MICA expression, whereas *MICA**002.01, characterized by distinct expression levels, exhibited a significant association with protection against the development of RA. Genetic variations within the MICA collectively contribute to the pathogenesis of RA.

Membrane MICA and sMICA are crucial for regulating the immune response to stimuli. Membrane-expressed MICA protein promotes the activation of NK and T cells. NK cell-mediated cytotoxicity and immunosuppressive sMICA could down-regulate NKG2D expression and encourage immunosuppressive CD4+ T-cell proliferation [39]. NKG2D-MICA engagement in SLE patients may initiate the mutual growth of MICA+ monocytes and NKG2D+CD4+ T cells [40]. However, sMICA promotes the proliferation of immunosuppressive NKG2D+CD4+ T cells, which correlates inversely with SLE disease activity [41]. A particular category of lupus individuals who had low levels of vitamin D, innate T-cell activation, and nephropathy had elevated levels of circulating sMICA [42], which were also substantially elevated among those with anti-SSB and anti-RNP autoantibodies [12]. Contact across MICA and NKG2D initiates NK cell activation and upregulation of CD69 and CD107 on NK cells, which contributes to NK cell exhaustion in SLE patients [12]. MICA and NKG2D have been identified as cytotoxic factors in cutaneous SLE [43]. The *MICA*-129Met/Val dimorphism influences MICA expression and the release of proteolytic sMICA [19]. Speedy NKG2D reduction on alloreactive CD8+T cells suggested that activation of *MICA*-129Met diminished the severity of acute graft-versus-host disease [20]. The combination of two of the three main markers (*DR*3-*DQ*2, *MICA*-5, and *MICA*-5.1) was related to an upsurge in genetic susceptibility and a reduced risk between *MICA*-9 and SLE [44], whereas the association between *MICA*-5.1 and SLE is owing to its disequilibrium of linkage with *HLA B**08 [45]. *HLA DRB1* *03, in conjunction with MICA-A5.1 and the absence of the two *MICA*-A6 and *HLA DRB**11 alleles, is strongly associated with SLE [46]. The *MICA* 129Met allele, *MICA*-A9 allele, and 129Met/Met genotype were all associated with SLE susceptibility, and the combination of 129Met/A-9 variants displayed reduced expression of NKG2D and NK cell cytotoxicity but increased IFN-γ production from NK92MI cells [47]. Han Chinese with *MICA**010 alleles had a reduced incidence of SLE [48]. The present study identified the *MICA**045 allele as the sole risk factor for SLE, while no association was observed between the *MICA**010 allele and SLE in Taiwanese subjects. The presence of 129Met in *MICA**045 (Appendix A), which mediates stronger NKG2D signals than 129Val, may partially explain the association between *MICA**045 and SLE. Nonetheless, the intricate MICA-NKG2D signaling pathways in various autoimmune inflammatory diseases have yet to be uncovered. 

Our study has limitations. The genetic interactions between *MICA* and *HLA* in the pathogenesis of autoimmune inflammatory diseases, particularly *HLA-C* in PSO and *HLA-DRB1* in RA, are an essential consideration [7]. Further, the effects of MICA alleles on modifications in serology tests, such as anti-cyclic citrullinated peptide (anti-CCP) positivity and clinical manifestations of autoimmune diseases, were not analyzed. Finally, we did not determine whether MICA alleles could affect outcomes of function-based therapies (especially biologic pharmaceuticals) in the treatment of PSO and RA [35,49]. 

## 4. Materials and Methods

### 4.1. Study Subjects 

Patients with psoriasis (N = 579), RA (N = 604), and SLE (N = 682) were recruited at the Chang Gung Memorial Hospital Rheumatology Clinic. Patients with psoriasis, RA, and SLE were diagnosed according to the respective diagnostic criteria for psoriasis, RA, and SLE. In the same geographical region, 928 healthy blood donors of similar age and gender were recruited as controls. The Chang Gung Memorial Hospital Institutional Review Board (Protocol No. 20200043B0) approved the human study, and all participants provided written consent for the study.

### 4.2. Determination of MICA Alleles by DNA Sequence Analysis

The sense primer (5′-CAA GAC CTT CCT TCC ACC ACC T-3′) and antisense primer (5′-CCT TGT CAC CAA CAT GCC TAT CTT T-3′) were utilized to amplify the *MICA* gene-specific DNA fragment (2352 base pairs) containing exons 2, 3, 4, and 5. Three sequencing primers (5′-CAG CAG ACC TGT GTG TTA A-3′, 5′-GGT GAT GGG TTC GGG AA-3′, and 5′-TTC CTC TCC CCT CCT TAG A-3′) and BigDye v3.1 Sequencing kit (Applied Biosystems, Foster City, CA, USA) were utilized in DNA sequence analyses of MICA exon 2, 3, and 4 on an ABI 3730xl DNA Analyzer. *MICA* DNA sequences were determined and *MICA* haplotypes or alleles were assigned according to IMGT/HLA database (ftp://ftp.ebi.ac.uk/pub/databases/ipd/imgt/hla/fasta/MICA_nuc.fasta, release date: 2022, accessed on 1 February 2023).

### 4.3. Construction of MICA Gene Expression Constructs

*MICA* cDNAs of peripheral blood mononuclear cells from the carriers of MICA alleles (*MICA**002, *MICA**008, *MICA**010, or *MICA**019) were amplified with RT-PCR and cloned into the lentiviral vector pCDH-CMV-EF1-copGFP (Systems Biosciences, Mountain View, CA, USA), as previously described [13].

### 4.4. Generation of Stable Cell Lines Expressing MICA Alleles

The C1R (ATCC#CRL-1573, Manassas, VA, USA) and LCL-721.221 (ATCC#CRL-1855) cell lines expressing empty vector, *MICA**002, *MICA**008, *MICA**010, or *MICA**019 alleles were gated as copGFP-positive and sorted on vFACSAria III cytometer (BD Biosciences, Mountain View, CA, USA), as previously described [13]. Stable cell lines were maintained in DMEM medium supplemented with 10% fetal calf serum and 1% GlutaMax (Invitrogen, Carlsbad, CA, USA) and incubated at 37 °C and 5% CO_2_. Prior to use, copGFP expressions were monitored by FACS.

### 4.5. Analysis of Membrane-Bound MICA by Flow Cytometry 

To determine surface expression of *MICA* alleles, the parental C1R and LCL-721.221 cells along with their respective stable clones expressing empty vector, *MICA**002, *MICA**008, *MICA**010, and *MICA**019 were stained separately with APC-conjugated anti-human MICA (clone 159227, R&D, Minneapolis, MN, USA) and MICA/MICB (clone 6D4, Biolegend, San Diego, CA, USA) for 30 min at 4 °C, followed by FACS analysis on an EC800 Flow Cytometry Analyzer (Sony Biotechnology, San Jose, CA USA). Transduced cells positive for GFP were gated for the determination of membrane-bound MICA.

### 4.6. Determination of MICA Trafficking

Total lysates of cells expressing MICA alleles were solubilized for 30 min on ice in lysis buffer (1% Triton X-100, 50 mM of Tris-Cl, pH 7.4, 300 mM of NaCl, 5 mM of EDTA, 0.02% NaN_3_) containing complete protease inhibitors (Roche Applied Science, Upper Bavaria, Germany). To investigate the trafficking of MICA alleles, the total cell lysates were deglycosylated with Endo H and PNGase F according to the manufacturer’s instructions (New England Biolabs, Ipswich, MA, USA). After the treatment, the samples were separated on 4–12% gradient SDS-PAGE, transferred to PVDF membranes (GE Healthcare, Uppsala, Sweden), and detected by rabbit anti-human MICA antibody (Abcam, Cambridge, UK) according to Methods section, as previously described [13].

### 4.7. NKG2D Binding Assay

LCL stable cell lines expressing vector control, *MICA**002, *MICA**008, *MICA**010, and *MICA**019 were incubated with recombinant human NKG2D-Fc fusion protein (R&D) at 4 °C or 37 °C, followed by staining with APC-conjugated anti-human IgG Fc. MICA surface expression was monitored by staining with anti-MICA/B (6D4, Biolegend) at 4 °C for 30 min, as described above. The binding capacity of rhNKG2D-Fc to MICA alleles was analyzed by EC800 Flow Cytometry Analyzer (Sony Biotechnology).

### 4.8. Determination of Soluble MICA (sMICA) Concentrations in Human Serum Samples

Concentrations of sMICA in serum samples from patients (psoriasis, RA, and SLE) and normal healthy controls were determined using a sandwich MICA DuoSet ELISA kit (R&D Systems, Minneapolis, MN, USA), according to the manufacturer’s instructions. The sMICA concentrations were calculated based on the standards from the same ELISA plate. Results were reported as the concentration of sMICA (pg/mL) in the serum sample.

### 4.9. Statistical Analysis

Logistic regression models were used to analyze relationship between the estimated haplotypes (alleles) and disease susceptibility. The 1% level of significance for *p*-values (*p* < 0.01) was used in genetic association analyses. Unpaired *t*-tests of GraphPad Prism 6.0 (GraphPad, La Jolla, CA, USA) were used to compare serum sMICA concentrations between normal healthy controls and PSA, RA, and SLE patients, respectively. The 5% level of significance (*p* < 0.05) was used for sMICA analysis.

## 5. Conclusions

Important functions are played by *MICA* variants in autoimmune inflammatory diseases. The *MICA**002 allele with high avidity protects against psoriasis and RA, whereas the deficient *MICA**010 allele increases the risk for psoriasis and RA development. The *MICA**045 containing high activity 129Met is associated with SLE susceptibility. The mechanisms underlying the effects of MICA alleles on the pathogenesis of autoimmune diseases may involve differences in MICA expression levels and avidity for NKG2D among *MICA* alleles. SLE disease activity might be affected by the elevated sMICA levels, which are influenced by the *MICA* alleles. Our data demonstrate that *MICA* variants impact the pathogenesis of common autoimmune inflammatory diseases, including PSO, RA, and SLE. The determination of *MICA* alleles could facilitate the diagnosis of autoimmune diathesis. 

## Figures and Tables

**Figure 1 ijms-25-03036-f001:**
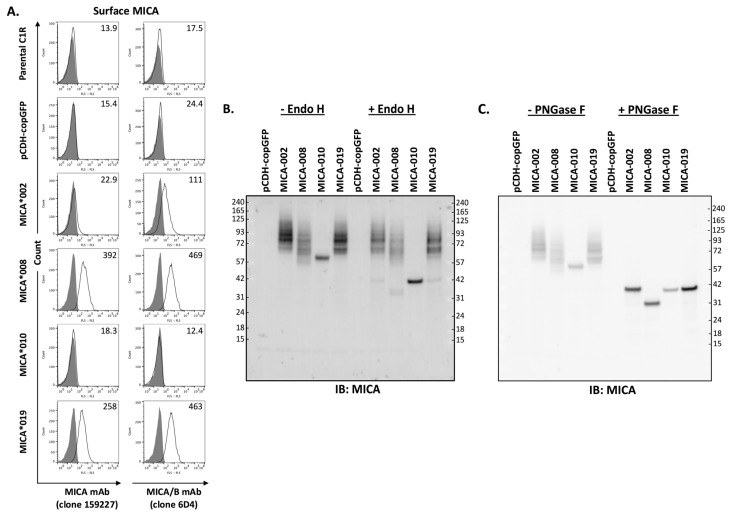
Analysis of MICA allele expression. (**A**). Three independent FACS analyses of surface expression levels of MICA alleles with APC-conjugated anti-MICA (clone 159227, left row panels) or anti-MICA/B (clone 6D4, right row panels) monoclonal antibodies (mAbs). Anti-MICA mAb (clone 159227, R&D, FAB1300A-100) detected a low level of cell surface MICA*002 in small percentage of cells, while anti-MICA/B mAb (clone 6D4, BioLegend, 320907) detected MICA*002 in majority of cells. Both mAbs detected MICA*008 and MICA*019 with similar capacity, while they failed to detect the expression of MICA*010 on cell surface. (**B**). Western blot analysis of MICA allele proteins treated by Endo H. The total lysates from C1R cells expressing vector pCDH-copGFP, MICA*002, MICA*008, MICA*010, or MICA*019 were untreated (−Endo H) or treated with Endo H (+Endo H) before immunoblot with anti-MICA antibodies, as described in “Materials and Methods”. Endo H treatment led to a significant decrease in MICA*010’s molecular weight, while the treatment did not cause molecular weight changes in most MICA*002, MICA*008, and MICA*019 proteins. (**C**). Western blot analysis of MICA alleles treated with PNGase F. PNGase F treatment led to similar molecular weights for MICA*002, MICA*010, and MICA*019. MICA*008 without transmembrane segment and cytoplasmic domain had a smaller molecular weight than other MICA alleles after PNGase F treatment. Both Western blots (**B**,**C**) are representatives of two independent experiments.

**Figure 2 ijms-25-03036-f002:**
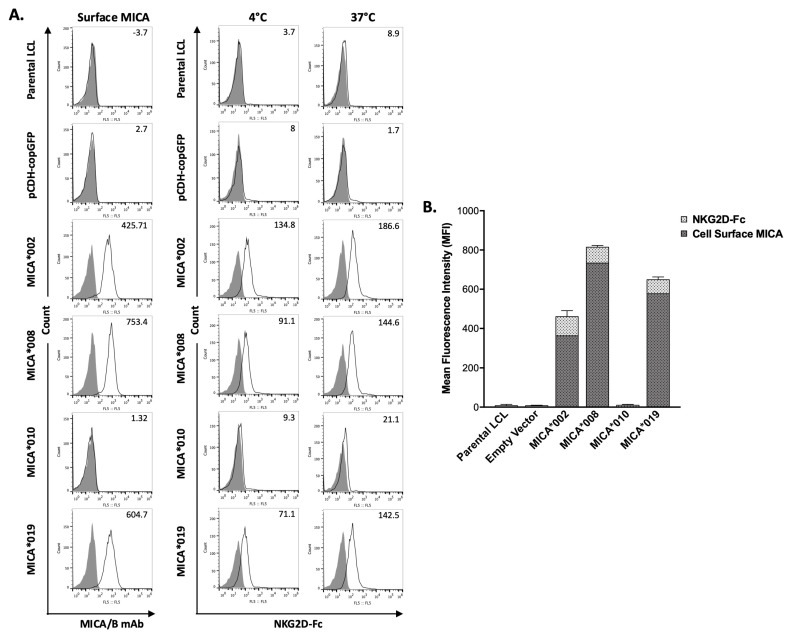
Binding capacity of MICA alleles to recombinant human NKG2D-Fc fusion protein (rhNKG2D-Fc). (**A**). FACS analysis of the binding of MICA alleles to rhNKG2D-Fc. MICA on LCL stable cell lines expressing vector control, MICA*002, MICA*008, MICA*010, and MICA*019 were detected by anti-MICA/B (6D4) (left column panels). Binding capacity of cell lines expressing MICA alleles at 4 °C (middle column panels) and 37 °C (right column panels) was determined by flow cytometry using rhNKG2D-Fc and APC-conjugated anti-human IgG Fc, as described in “Materials and Methods”. Representative histograms of the cell surface staining of isotype controls (filled) and MICA (black lines) with the MFI values of MICA are shown (left column panels). The staining of the cells with APC-conjugated anti-human IgG Fc was used as negative control (filled). The binding of human rhNKG2D-Fc fusion protein to the cells at 4 °C (middle column panels) and 37 °C (right column panels) was detected by APC-conjugated anti-human IgG Fc (black lines). The MFI (mean fluorescent intensity) values of rhNKG2D-Fc surface staining are indicated. (**B**). Bar graph showing the average MFI of three independent experiments. The MFI values of MICA staining are shown with dark grey bars indicating the expression levels of different surface MICA alleles; MFI values of rhNKG2D-Fc staining are shown with light grey bars representing the binding ability of NKG2D to the respective surface MICA. MICA*008 expression level on cell surface was highest among MICA alleles. MICA*019 expression level was higher than MICA*002. MICA*002 had highest binding capacity among MICA variants. Cells expressing MICA*010 did not express surface MICA and had background staining by rhNKG2D-Fc. MICA*008 and MICA*019 had similar binding capacity at 4 °C and 37 °C.

**Figure 3 ijms-25-03036-f003:**
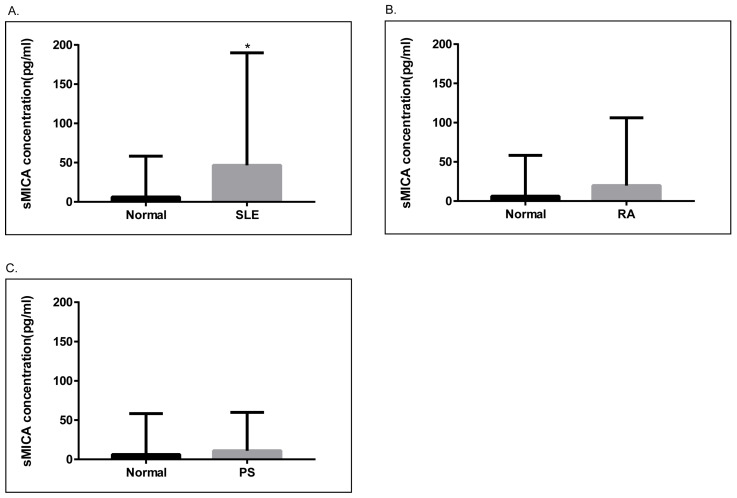
Analyses of soluble MICA (sMICA) in psoriasis, RA, and SLE patients. (**A**). The concentrations of sMICA were significantly higher (*p* = 0.01) in SLE patients (N = 139, 46.66 ± 12.15 pg/mL) as compared to healthy controls (N = 92, 6.224 ± 5.417 pg/mL). (**B**). The concentrations of sMICA were not significantly different between healthy controls and RA patients (N = 80, 18.42 ± 10.55 pg/mL). (**C**). The concentrations of sMICA were not significantly different between healthy controls and psoriasis patients (N = 63, 11.08 ± 5.461 pg/mL). *, *p* < 0.05.

**Table 1 ijms-25-03036-t001:** Association of MICA alleles with PSO in Taiwanese subjects.

MICA Alleles	Estimated Frequency			Logistic Regression		Logistic Regression Adjusted for Sex	
	PSO	Normal	All	*p* Value	OR (95% CI)	P_FDR_ Value	OR (95% CI)
	(2N = 1158)	(2N = 1858)	(2N = 3016)				
MICA*010:01	29.97% (347)	16.68% (310)	21.78% (657)	4.01 × 10^−17^	2.19 (1.82–2.63)	1.93 × 10^−15^	2.18 (1.82–2.62)
MICA*008:01	28.50% (330)	30.95% (575)	30.01% (905)	0.1563	0.89 (0.76–1.05)	0.9658	0.89 (0.76–1.04)
MICA*002:01	16.58% (192)	20.56% (382)	19.03% (574)	0.0072	0.77 (0.64–0.93)	0.0666	0.77 (0.64–0.93)
MICA*019:01	7.60% (88)	7.97% (148)	7.82% (236)	0.7179	0.95 (0.72–1.25)	0.9999	0.95 (0.73–1.25)
MICA*012:01	5.70% (66)	8.40% (156)	7.36% (222)	0.0064	0.66 (0.49–0.89)	0.0638	0.66 (0.49–0.89)
MICA*045	5.27% (61)	5.11% (95)	5.17% (156)	0.8520	1.03 (0.74–1.44)	0.9999	1.05 (0.76–1.47)
MICA*009:01	3.71% (43)	7.05% (131)	5.77% (174)	0.0003	0.53 (0.37–0.75)	0.0058	0.53 (0.38–0.75)
MICA*004	0.52% (6)	0.86% (16)	0.73% (22)	0.3071	0.62 (0.25–1.55)	0.9999	0.61 (0.24–1.52)

**Table 2 ijms-25-03036-t002:** Association of *MICA* alleles with RA in Taiwanese subjects.

MICA Alleles	Estimated Frequency			Logistic Regression		Logistic Regression Adjusted for Sex	
	RA (2N = 1208)	Control (2N = 1858)	All	P_FDR_ Value	OR (95% CI)	P_FDR_ Value	OR (95% CI)
MICA*008:01	30.55% (369)	30.95% (575)	30.79% (944)	0.8158	0.98 (0.84–1.15)	0.1041	0.84 (0.69–1.04)
MICA*010:01	22.52% (272)	16.68% (310)	18.98% (582)	5.03 × 10^−5^	1.47 (1.22–1.77)	0.0011	1.51 (1.18–1.93)
MICA*002:01	11.75% (142)	20.56% (382)	17.09% (524)	1.08 × 10^−9^	0.52 (0.43–0.64)	4.16 × 10^−6^	0.53 (0.41–0.70)
MICA*012:01	11.75% (142)	8.40% (156)	9.72% (298)	0.00278	1.44 (1.13–1.82)	0.0330	1.41 (1.03–1.93)
MICA*019:01	8.44% (102)	7.97% (148)	8.15% (250)	0.6347	1.07 (0.82–1.39)	0.1586	1.29 (0.91–1.84)
MICA*045	5.96% (72)	5.11% (95)	5.45% (167)	0.3113	1.18 (0.86–1.62)	0.1589	1.35 (0.89–2.05)
MICA*009:01	5.30% (64)	7.05% (131)	6.36% (195)	0.0593	0.75 (0.55–1.01)	0.9867	1.00 (0.68–1.47)
MICA*001	0.83% (10)	0.11% (2)	0.39% (12)	0.0149	6.52 (1.44–29.49)	0.0080	12.44 (1.93–80.19)
MICA*004	0.66% (8)	0.86% (16)	0.78% (24)	0.5575	0.78 (0.34–1.78)	0.0694	0.37 (0.12–1.08)
MICA*008:02	0.41% (5)	0.22% (4)	0.29% (9)	0.3284	1.93 (0.52–7.22)	0.6333	1.48 (0.30–7.41)
MICA*072	0.33% (4)	0.11% (2)	0.20% (6)	0.1935	3.09 (0.56–16.92)	0.8524	0.83 (0.12–5.72)

**Table 3 ijms-25-03036-t003:** Association of *MICA* alleles with SLE in Taiwanese subjects.

MICA Alleles	Estimated Frequency			Logistic Regression		Logistic Regression Adjusted for Sex	
	SLE (2N = 1364)	Control (2N = 1858)	All	P_FDR_ Value	OR (95% CI)	P_FDR_ Value	OR (95% CI)
MICA*008:01	30.13% (411)	30.95% (575)	30.60% (986)	0.6209	0.96 (0.83–1.12)	0.0598	0.81 (0.64–1.01)
MICA*002:01	19.57% (267)	20.56% (382)	20.14% (649)	0.5005	0.94 (0.79–1.12)	0.3699	1.13 (0.87–1.46)
MICA*010:01	16.13% (220)	16.68% (310)	16.45% (530)	0.6703	0.96 (0.79–1.16)	0.9243	0.99 (0.74–1.31)
MICA*019:01	9.02% (123)	7.97% (148)	8.41% (271)	0.2912	1.14 (0.89–1.47)	0.2151	1.27 (0.87–1.84)
MICA*045	8.58% (117)	5.11% (95)	6.58% (212)	0.0001	1.74 (1.31–2.30)	0.0002	2.24 (1.47–3.43)
MICA*012:01	7.11% (97)	8.40% (156)	7.85% (253)	0.1904	0.84 (0.65–1.09)	0.0670	0.71 (0.49–1.03)
MICA*009:01	6.52% (89)	7.05% (131)	6.83% (220)	0.5637	0.92 (0.70–1.21)	0.6874	1.09 (0.72–1.64)
MICA*004	0.51% (7)	0.86% (16)	0.71% (23)	0.2709	0.62 (0.26–1.46)	0.3758	0.60 (0.19–1.87)
MICA*033	0.37% (5)	0.11% (2)	0.22% (7)	0.1421	3.42 (0.66–17.70)	0.3014	3.54 (0.32–38.91)
MICA*002:03	0.29% (4)	0.00% (0)	0.12% (4)				
MICA*028	0.29% (4)	0.00% (0)	0.12% (4)				

## Data Availability

The data presented in this study are available on request from the corresponding author.

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
