# Peer review of "Functional *MICA* Variants Are Differentially Associated with Immune-Mediated Inflammatory Diseases"

_ijms, 2024, doi:10.3390/ijms25053036_

Round 1

Reviewer 1 Report

Comments and Suggestions for Authors

Rheumatoid arthritis, psoriasis, and systemic lupus erythematosus are chronic inflammatory diseases with an autoimmune basis. The aetiology of these diseases is multifactorial and so far remains unclear. It is presumed that a genetic background is largely responsible for the risk of their development.

The findings presented here contribute to the current state of knowledge on the risk factors for these diseases in Taiwanese patients.

Please find below my comments.

1.The manuscript was not prepared using the IJMS LaTeX template files

2. All Figures, Schemes, and Tables should be inserted into the main text close to their first citation

3. No citation of figures 1B and 1C in the chapter Results

4. Editing errors in text footers and headers

5. Not very clear figures 1 and 2: axis descriptions not visible

6. In conclusion, please add information on the potential for practical use of the assessment of the polymorphism of the MICA molecule in the treatment of the autoimmune diseases discussed in the paper.

Author Response

Dear editor and reviewers :

We appreciate the comments from the reviewers. We would like to offer our point-by point responses. With these revisions, we hope our manuscript would be acceptable for publication.

Reviewer 1:

  1. The manuscript was not prepared using the IJMS LaTeX template files.

Response: We revised our manuscript based on the provided IJMS official format.

  1. All Figures, Schemes, and Tables should be inserted into the main text close to their first citation

Response: All Figures and Tables are placed close to their first citation in the main text in the revised manuscript.

  1. No citation of figures 1B and 1C in the chapter Results

Response: We corrected our errors and the citations of figures 1B and 1C are now included in the main text of the “Results” section on page 6.

  1. Editing errors in text footers and headers

Response: We extensively revised our manuscript and corrected our errors in the manuscript.

  1. Not very clear figures 1 and 2: axis descriptions not visible

Response: For clarification, we inserted representative X and Y axis with enlarged titles onto each histogram.

.

  1. In conclusion, please add information on the potential for practical use of the assessment of the polymorphism of the MICA molecule in the treatment of the autoimmune diseases discussed in the paper.

Response: We added the sentences “SLE disease activity might be affected by the elevated sMICA levels, which are influenced by the MICA alleles. Our data demonstrate that MICA variants impact pathogenesis of common autoimmune inflammatory diseases including PSO, RA, and SLE. Determination of MICA alleles could facilitate the diagnosis of autoimmune diathesis.” in the “Conclusion” section on page 11 for the potential applications of our findings.

Reviewer 2 Report

Comments and Suggestions for Authors

I reviewed the manuscript “Functional MICA variants are differentially associated with immune mediated inflammatory diseases” has been reviewed.

The study evaluated the impact of MICA variations on the development of psoriasis (PSO), rheumatoid arthritis (RA), and systemic lupus erythematosus.

The introduction and rationale for the study are well written. However, the latest pertinent literature should be added wherever it is appropriate.

In general, the in vitro cell culture studies are repeated for obtaining effective results? Are these studies in the manuscript triplicated or repeated? If so, please mention in the manuscript.

The limitation of the study has to be added.

Comments on the Quality of English Language

Minor grammar and language revision is required 

Author Response

Reviewer 2:

1 The introduction and rationale for the study are well written. However, the latest pertinent literature should be added wherever it is appropriate.

Response:  We updated the introduction to incorporate a more current and thorough review and revised the limits section to feature new references that should encourage further study.

  1. In general, the in vitro cell culture studies are repeated for obtaining effective results? Are these studies in the manuscript triplicated or repeated? If so, please mention in the manuscript.

Response: All experiments were independently either duplicate or triplicate, which was described in the figure legends.

3 The limitation of the study has to be added.

Response: We add a paragraph to describe the limitations of our study. 

Round 2

Reviewer 1 Report

Comments and Suggestions for Authors

IJMS] Manuscript ID: ijms-2851982 - Revised Review

The authors have addressed and made factual corrections, while editorial errors have not been satisfactorily corrected, i.e. :

1. The manuscript was not prepared using the IJMS LaTeX template files – available in Instructions for Authors (…..Use the Microsoft Word template or LaTeX template to prepare your manuscript…)

3. References are not prepared in accordance with the requirements of the IJMS journal (see; Instructions for Authors):

References should be described as follows:

  • Journal Articles:
    1. Author 1, A.B.; Author 2, C.D. Title of the article. Abbreviated Journal Name Year, Volume, page range.